# GENERALIZED REPRESENTATION FOR GENERALIZED DYNAMICS GENERATION

## ABSTRACT

Digital twin worlds with realistic interactive dynamics presents a new opportunity to develop generalist embodied agents in scannable environments with complex physical behaviors. To this end, we present **Gen-3** (**Gen**eralized Representation for **Gen**eralized Dynamics **Gen**eration), a framework that takes a potential energy perspective to seamlessly integrate rigid body, articulated body, and soft body dynamics into a unified, geometry-agnostic system. Gen-3 operates from the governing principle that the potential energy for any stable physical system should be low. This fresh perspective allows us to treat the world as one holistic entity and infer underlying physical properties from simple motion observations. We extend classic elastodynamics by introducing directional stiffness to capture a broad spectrum of physical behaviors, covering soft elastic, articulated, and rigid body systems. We propose a specialized network to model the extended material property and employ a neural field to represent deformation in a geometry-agnostic manner. Extensive experiments demonstrate that Gen-3 robustly unifies diverse simulation paradigms, offering a versatile foundation for creating interactive virtual environments and training robotic agents in complex, dynamically rich scenarios. [1]

## 1 INTRODUCTION

Scannable digital twin worlds with realistic physical interactive dynamics is crucial in training generalist embodied agents to help them develop skills dealing with diverse real-world scenarios. The demand for simple ways to creating digital twin worlds is particularly driven by advancements in robotics and mixed reality. These applications seek for a method that can not only replicate complex physical behaviors but also allow for seamless interaction with diverse objects. However, traditional physics simulations are often fragmented into rigid body, articulated body, and soft body simulations, with each category making different assumptions and following its own paradigm.

One critical step towards generating such complex dynamics is to devise a unified framework that seamlessly integrates these multiple simulation paradigms. This paper brings a fresh perspective, treating the world as one holistic entity and modeling its diverse dynamics with as few assumptions as possible. However, developing such a system presents substantial challenges: first, defining an integrated mathematical formulation that unifies the physical assumptions across different dynamics types, *e.g.* soft elastic, articulated, and rigid scene; second, handling the heterogeneity of geometric representations, where rigid and articulated bodies might use meshes or implicit functions, while soft bodies use point clouds, Gaussian splats, or particle fields. Our work tackles these challenges with a unified framework that enables more versatile, physically simulatable assets and environments, as shown in Figure 1. These assets are foundational for creating interactive virtual twins with diverse physical behaviors and for providing effective world models for training robotic agents to learn and adapt in complex scenarios.

In this paper, we introduce **Gen-3**, a unified framework capable of describing diverse physical behaviors, including soft elastic body, rigid body, and articulated body dynamics (an overview is provided in Figure 2). The governing principle to our proposed approach lies in the low potential energy states for any stable physical systems. That is, any observed dynamics should induce relatively low energy compared to physically unrealistic dynamics. This perspective allows us to cast our

---

[1]More visual results on our anonymous page: https://sites.google.com/view/iclr26anonymoussubmission/home?authuser=3.

Table 1: Compared to existing methods, our Gen-3 method enables different simulation capabilities.

| Algorithms | Geometry | Soft Body | Articulated | Discontinuum |
|---|---|---|---|---|
| Simplicits (Modi et al., 2024) | ✓ | ✓ | | |
| PhysGaussian (Xie et al., 2023) | | ✓ | | |
| PhysDreamer (Zhang et al., 2024b) | | ✓ | | |
| SpringGauss (Zhong et al., 2024) | | ✓ | | |
| Constitutive (Lin et al., 2025) | | ✓ | | |
| ArticulateAnything (Le et al., 2024) | | | ✓ | |
| URDFormer (Chen et al., 2024) | | | ✓ | |
| PhysGen (Liu et al., 2024b) | | | | ✓ |
| Gen-3 (Ours) | ✓ | ✓ | ✓ | ✓ |

Figure 1: Gen-3 handles diverse dynamics and geometry representation.

problem into elastodynamics, using the elasticity energy function to enable interaction. To achieve a universal formulation, we introduce directional stiffness via anisotropic Young's modulus, a material parameter that allows us to capture a broad spectrum of physical behaviors. This includes soft elasticity, articulated motion, and near-rigid responses, where apparent discontinuum (like object separation) can emerge from learned localized stiffness variations, effectively approaching near-zero stiffness in specific regions. We introduce an energy-based contrastive training scheme to train a specialized network that predicts isotropic and anisotropic stiffness fields in $\mathbb{R}^4$ over the domain of interest. To handle heterogeneous representations, we model deformation as a set of sparse linear weights represented as a $\mathbb{R}^3$ neural field over the object domain, which is also called motion eigenmodes (Modi et al., 2024; Benchekroun et al., 2023). Consequently, Gen-3 is geometry-agnostic and robustly unifies diverse simulation paradigms within a single framework. In summary, our contributions are as follows:

- We introduce anisotropic material stiffness to enable simulation of diverse dynamics, including soft body, rigid body, and articulated body systems, all within a unified potential energy perspective.

- We devise an energy-based training scheme and effective network modules to model deformation to be geometry representation agnostic and predict material properties from single observed trajectory.

- We show the universality of Gen-3 in unifying diverse simulation paradigms and geometry representations through extensive experiments, offering a versatile foundation for interactive virtual environments and robotic agent training experiments.

## 2 RELATED WORK

**4D Generation and Dynamic 3D Simulation** Most efforts in 4D generation and dynamic 3D simulation leverage text-to-image and video diffusion models, particularly Score Distillation Sampling (SDS)(Poole et al., 2023), using diverse dynamic 3D representations such as HexPlane(Singer et al., 2023), multi-scale 4D grids (Zhao et al., 2023), K-plane (Jiang et al., 2024b), multi-resolution hash encoding (Bahmani et al., 2024), disentangled canonical NeRF (Zheng et al., 2024), 3D Gaussians with deformation fields (Ling et al., 2024), warped Gaussian surfels (Wang et al., 2024), Nerfies (Park et al., 2021a), Dynamic 3D Gaussians (Luiten et al., 2023), and 4D Gaussians (Duan et al., 2024). Recent studies (Zhang et al., 2024a; Jiang et al., 2024a) emphasize multi-view video generative models to ensure spatial-temporal consistency by providing improved gradients during distillation. Alternative strategies include video-first generation for direct appearance and motion reference in optimizing 3D representations (Ren et al., 2023; Yin et al., 2023b; Pan et al., 2024; Zeng et al., 2024), or utilizing generalizable reconstruction methods to expedite the process (Ren et al., 2024). While these approaches achieve high visual fidelity and geometric coherence, they typically neglect explicit physical dynamics modeling, thereby limiting their effectiveness in novel scenarios demanding realistic physical and material behaviors (Li et al., 2008; Newcombe et al., 2015; Attal et al., 2023; Li et al., 2023b; Pumarola et al., 2021; Park et al., 2021b). SCD is a numerical integration framework that unifies elastic and constrained models at the solver level for a given object, assuming geometry and material parameters are known. In contrast, Gen-3 is a learning-based system identification method that infers neural stiffness fields (including anisotropic Young's modulus) and deformation eigenmodes directly from motion. We scope our work under physical 4D dynamics generation and extends beyond the traditional elastodynamics by introducing new physics parameter term directional stiffness, named anisotropic Young's Modulus. This new term allows us to model diverse dynamic behaviors, from soft elastodynamics, to rigid body and articulated systems. Other traditional methods such as stable constrained dynamics (Tournier et al., 2015) unifies elastic and constrained models

from a solver perspective, for a given object, assuming geometry and material parameters are known. In contrast, Gen-3 is a learning-based system identification method that infers neural stiffness fields (including anisotropic Young's modulus) and deformation eigenmodes directly from motion.

**Interactive Dynamics Generation** Prior research has explored generating interactive dynamics in both 2D and 3D content based on user-specified preferences or constraints. For animating images, initial conditions such as driving videos (Siarohin et al., 2019a;b; 2021; Karras et al., 2023), keypoint trajectories (Hao et al., 2018; Blattmann et al., 2021; Chen et al., 2023a; Yin et al., 2023a; Li et al., 2024), or textual prompts (Ho et al., 2022; Yang et al., 2023; Chen et al., 2023b;c; Zhang et al., 2023) have guided the process. Recent works including WonderPlay (Li et al., 2025) uses keypoint trajectory and inductive priors to determine appropriate physics simulator to use to simulate behaviors in a 2D scene. Our work pursues a completely different philosophy. By removing assumptions and priors, we aim to directly learn the underlying physical system the from observed behaviors in a unified physical system. In this way, we are freed from limitations and mismatches of physical priors. Recent advances have extended these interactive methods to 3D scenarios (Jiang et al., 2024a; Ling et al., 2024). To maintain physically plausible dynamics, several studies (Li et al., 2023a; Qiu et al., 2024; Borycki et al., 2024; Zhong et al., 2024; Fu et al., 2024; Feng et al., 2024) have incorporated physical constraints. Specifically, PhysGaussian (Xie et al., 2024) combines Gaussian representations with Material Point Method (MPM) simulations. However, current 3D representations lack inherent material characteristics, forcing manual specification of material properties per particle. Subsequent works (Huang et al., 2024; Zhang et al., 2024b; Liu et al., 2024a; Lin et al., 2025) have sought to automatically learn these physical parameters using diffusion models. Other works leverages differentiable MPM for identifying system parameters. Specifically, NCLaw (Ma et al., 2023), NeuMA (Cao et al., 2024), Omniphys4D (Lin et al., 2025) proposes using elasticity and plasticity law to regress the parameters to generate desired motion. GausSim (Shao et al., 2024) proposes a hierarical grouping in the Gaussian space to speed up simulation. However, these works only considers soft elastic objects, assuming consistent soft material for the simulated shape. On the other hand, we go beyond standard elastodynamics by introducing a new physical parameter, directional stiffness (anisotropic Young's modulus). This additional term enables us to capture a wide spectrum of dynamic behaviors, ranging from soft elastic materials to rigid bodies and articulated systems. Other works focusing only on rigid body simulation systems (Le et al., 2024; Chen et al., 2021; Yu et al., 2025) by first segmenting the shape, inferring the joint type from a set of predefined joints, and then applying rigging or skinning for interative dynamics. These works only works with articulated shapes, such as furnitures, and they do not consider other types of dynamics such as soft elastic dynamics or discontinuum dynamics. Our work pursues a completely different philosophy, we aim to reduce as much assumption as possible, governing the by one physics principle to model and reduce energy potential, instead of leveraging inductive bias or LLM or diffusion model priors.

## 3 PROBLEM STATEMENT AND PHYSICS PRELIMINARIES

**Problem Statement.** In this work, we aim to unify diverse, commonly observed dynamics in one physics-based framework. Given motion observations $\{\mathbf{x}_t\}_{t=1}^T$, our task is to infer the underlying physical system that governs the behavior of the geometry $\mathbf{x}^{\text{rest}}$. With the influence free input geometry $\mathbf{x}^{\text{rest}}$, we can then extrapolate to various types of plausible dynamics under new interactions.

When given an input trajectory, denoted as $\mathbf{x}_{t=0}^T$, our goal is to make the initial geometric configuration $\mathbf{x}^{\text{rest}}$ physically simulatable and respect the physical system underlying observed behavior $\mathbf{x}_{t=0}^T$. When new physical influences are imposed, *e.g.*, new interactions or forces, the geometric domain of interest $\mathbf{x}^{\text{rest}}$ extrapolates to new dynamics and behaves in physically plausible ways.

**Physics Preliminary.** To construct a unified framework that accommodates various dynamics, we leverage the classical elasticity energy function and extend it to accommodate common motions, including soft-elastic, rigid-nondeforming, and articulated dynamics. Continuum mechanics, governed by Hook's law $\sigma = \mathbf{E}\Phi$, states the relationship between stress $\sigma$ and strain $\Phi$ by the material stiffness $\mathbf{E}$. In its commonly used form, the Neohookean elasticity energy potential (Kim & Eberle, 2020; Sifakis & Barbic, 2012), the deformation of materials is described by a mapping $\phi$ that takes a point from the original, undeformed configuration $\mathbf{x}$ to its new location in the deformed configuration $\mathbf{x}_t = \phi(\mathbf{x}, t)$. The derivative of this mapping with respect to $\mathbf{x}$ is the deformation gradient $\mathbf{F} = \nabla_{\mathbf{x}}\phi(\mathbf{x}, t)$. This tensor is essential for formulating how stress relates to strain in Neohookean

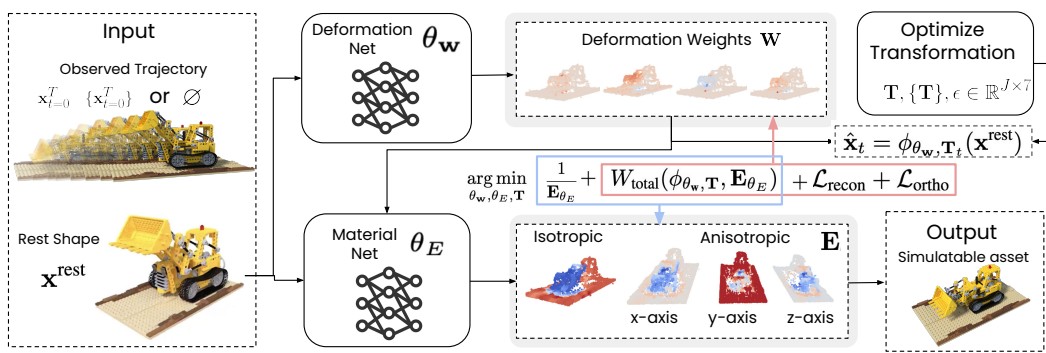

Figure 2: **Overview of the Gen-3 algorithm**: Gen-3 takes in a geometry $\mathbf{x}^{\text{rest}}$, and if trajectories are given, we reconstruct the observed motion $\{\mathbf{x}_t\}_t$ by optimizing both motion eigenmode weights $\mathbf{w}$ (output of the deformation net $f_{\theta_{\mathbf{w}}}$) and the transformation handles $\mathbf{T}$. We then use the predicted eigenmodes weights $\mathbf{w}$ and transformations $\mathbf{T}$ to calculate elasticity quantities, including deformation gradient $\mathbf{F}$, to be fed into the Young's modulus net $f_{\theta_{\mathbf{E}}}$ to predict the isotropic and anisotropic Young's Moduli $\mathbf{E}$. The predictions including motion eigenmode weights and Young's Moduli can be used to simulate new motion dynamics (right bottom). We mark the training objective of $f_{\theta_{\mathbf{w}}}$ in red color and objectives of $f_{\theta_{\mathbf{E}}}$ in blue color.

elasticity energy, which takes the form of

$$W = \frac{\mu}{2} \left( \text{tr}(\mathbf{C}) - 3 \right) + \frac{\lambda}{2} \left( \det(\mathbf{F}) - 1 \right)^2, \tag{1}$$

where $\mathbf{C}$ is the Cauchy stress tensor, defined as $\mathbf{C} = \mathbf{F}^{\top}\mathbf{F}$ (Kim & Eberle, 2020). Lamé parameters $\mu$ and $\lambda$ are related to classical isotropic material stiffness Young's modulus $E \in \mathbb{R}$ and Poisson's ratio $\nu \in \mathbb{R}$ via $\mu = \frac{E}{2(1+\nu)}, \quad \lambda = \frac{E\nu}{(1+\nu)(1-2\nu)}$. The Neohookean energy equation 1 states that the potential energy of any physical entity depends on the material parameters $E$ and $\nu$ as well as the deformation map $\phi$, which then defines $\mathbf{F}$ and $\mathbf{C}$.

## 4 METHODOLOGY

The goal of our work is to uncover these underlying physical quantities from observed dynamics based on the following governing principle:

> *The potential energy for any stable conservative physical system should be low.*

In Section 4.1, we first describe how we leverage motion observations to obtain a differentiable deformation map $\phi(\cdot)$. In Section 4.2, we define our total energy $W$, and find material parameters by minimizing our proposed NeoHookean energy following our governing principle.

### 4.1 LEARNING DEFORMATION FUNCTION FROM MOTION OBSERVATION

**Reduced-order Dynamics.** Reduced-order simulation (Benchekroun et al., 2023; Fulton et al., 2019; Modi et al., 2024; Sharp et al., 2023; Chen et al., 2022) leverage efficient representations of deformable bodies to enable fast simulation performance. These methods approximate high-resolution deformations using low-dimensional subspaces, such as complementary eigenmodes (Benchekroun et al., 2023), compact latents (Chen et al., 2022), or cubature points (Fulton et al., 2019; Modi et al., 2024). Following the convention of (Benchekroun et al., 2023; Modi et al., 2024), for each point $x_i^{\text{rest}} \in \mathbb{R}^3, 1 \le i \le N$ of the rest geometry $\mathbf{x}^{\text{rest}} \in \mathbb{R}^{N \times 3}$, where $N$ is the number of points in $\mathbf{x}^{\text{rest}}$, we use a set of linear weights $\mathbf{w} \in \mathbb{R}^{N \times J} : \{w_{i,j}\}_{j=1}^{J}$ to combine a sparse set of $J$ transformation handles $\mathbf{T}_{j,t} \in \mathbb{R}^{3 \times 4}$ to map a rest geometry $\mathbf{x}^{\text{rest}}$ to a deformed state $\mathbf{x}_t \in \mathbb{R}^{N \times 3} := \{x_{i,t}\}_{i=1}^{N}$ at timestep $t$, using a deformation function $\phi$. We call these weights $\mathbf{w}$ to be eigenmodes.

$$x_{i,t} = \phi(x_i^{\text{rest}}, t) \triangleq \sum_{j=1}^{J} w_{i,j} \mathbf{T}_{j,t} x_0^{\text{rest}}. \tag{2}$$

The reduced-order property arises from the compactness of the system since $J \ll N$.

**Neural Deformation Eigenmodes.** We want our model to be agnostic of specific geometric representations and applicable to data with any number of points $N$. We require the parameterization of

---

**Algorithm 1** Gen-3 algorithm

---

**Require:** Rest geometry $\mathbf{x}^{\text{rest}} \in \mathbb{R}^{N \times 3}$, observed trajectory $\{\mathbf{x}_t\}_{t=0}^T$ $\left(\mathbf{x}_t \in \mathbb{R}^{N \times 3}\right)$

**Ensure:** Physical stiffness parameter $\mathbf{E}$, trained deformation net $\theta_{\mathbf{w}}$, material net $\theta_{\mathbf{E}}$, optimized handle transformations $\mathbf{T} = \{\mathbf{T}_t\}_{t=0}^T$.

    *// Initialization*

1: Initialize deformation MLP $f_{\theta_{\mathbf{w}}}$ (outputs $\mathbf{w} \in \mathbb{R}^{N \times J}$).
2: Initialize material MLP $f_{\theta_{\mathbf{E}}}$ (outputs $\mathbf{E} \in \mathbb{R}^{N \times 4}$: $[E_{\text{iso}}; E_{\text{aniso},k}], k = 1, 2, 3$).
3: Initialize SE(3) handle transformations $\mathbf{T}_t \in \mathbb{R}^{J \times 7}, \forall t$ (quaternion + translation).

    *// Training*

4: **while** training not converged **do**
5:     Reconstruct observed trajectory $\{\mathbf{x}_t\}_{t=0}^T$ with $\theta_{\mathbf{w}}, \mathbf{T}$ as $\hat{\mathbf{x}}_t = \phi_{\theta_{\mathbf{w}}, \mathbf{T}_t}(\mathbf{x}^{\text{rest}})$ from Equation 3
6:     Calculate $\mathbf{F}$ and $\mathbf{C}$ from $\phi_{\theta_{\mathbf{w}}, \mathbf{T}_t}(\mathbf{x}^{\text{rest}})$ Equation 6.
7:     Obtain $\mathbf{E}$ parameterized by Network $\theta_{\mathbf{E}}$ from Equation 9.
8:     Calculate $W_{\text{total}}$ with $\mathbf{F}, \mathbf{C}$ and $\mathbf{E}$ from Equation 8.
9:     Supervise with complete training loss $\mathcal{L}_{\text{ortho}}, \mathcal{L}_{\text{recon}}, W_{\text{total}}$ and $\mathbf{E}$ from Equation 11.
10:    Update $\theta_{\mathbf{w}}, \theta_{\mathbf{E}}, \mathbf{T}$ by optimizing the training loss.

---

the eigenmodes $\mathbf{w}$ to be differentiable to compute the deformation gradient $\mathbf{F}$ and the Cauchy stress tensor $\mathbf{C}$ (see Equation 6). Therefore, we consider using a neural network. We thus choose to use a neural field that models the volume in 3D space to characterize the observed motion dynamics (Modi et al., 2024; Benchekroun et al., 2023; Kim & Eberle, 2020). The neural field $f$, parameterized by an MLP with parameters $\theta_{\mathbf{w}}$, predicts eigenmode weights $\mathbf{w} = f(\mathbf{x}^{\text{rest}}; \theta_{\mathbf{w}})$. The deformation map $\phi$ is then a reduced-order model parameterized by $\theta_{\mathbf{w}}$ and transformation handles $\mathbf{T}$. Therefore, the predicted deformed geometry $\hat{\mathbf{x}}_t$ under neural eigenmodes field is

$$\hat{\mathbf{x}}_t = \phi_{\theta_{\mathbf{w}}, \mathbf{T}_t}(\mathbf{x}^{\text{rest}}) = f(\mathbf{x}^{\text{rest}}; \theta_{\mathbf{w}}) \mathbf{T}_t \mathbf{x}^{\text{rest}}. \tag{3}$$

For both the compactness of complementary dynamics and the invertibility of deformations, we need to enforce orthogonality of the learned eigenmodes. We train the eigenmode field with orthogonality regularization $\mathcal{L}_{\text{ortho}}$ following (Modi et al., 2024), where $j, j'$ are handle indices, and $\delta_{j,j'}$ is the Kronecker delta:

$$\mathcal{L}_{\text{ortho}}(\theta_{\mathbf{w}}) = \frac{1}{J^2} \sum_{j=1}^J \sum_{j'=1}^J \left( f(\mathbf{x}^{\text{rest}}; \theta_{\mathbf{w}})_j^\top f(\mathbf{x}^{\text{rest}}; \theta_{\mathbf{w}})_{j'} - \delta_{j,j'} \right)^2, \tag{4}$$

which makes sure we obtain a complementary subspace on eigenmodes $\mathbf{w}$.

**Learning Neural Eigenmodes as Deformation Reconstruction.** We use a reconstruction loss between predicted $\hat{\mathbf{x}}_t$ from Equation 3 and observed deformations $\mathbf{x}_t$ to supervisedly train $\theta_{\mathbf{w}}$ and $\mathbf{T}$:

$$\mathcal{L}_{\text{recon}}(\theta_{\mathbf{w}}, \mathbf{T}) = \sum_t \mathcal{D}(\mathbf{x}_t, \hat{\mathbf{x}}_t) = \sum_t \mathcal{D}(\mathbf{x}_t, f(\mathbf{x}^{\text{rest}}; \theta_{\mathbf{w}}) \mathbf{T}_t \mathbf{x}^{\text{rest}}), \tag{5}$$

where $\mathcal{D}$ is a distance metric. If the trajectory is 3D points tracked over time, we use the L2 distance $\mathcal{D}_{\text{L2}}(x, y) = \|x - y\|_2^2$; otherwise, we alternatively use the Chamfer distance $\mathcal{D}_{\text{Chamfer}}(x, y) = \sum_{x_i \in x} \min_{y_j \in y} \|x_i - y_i\|_2^2 + \sum_{y_j \in y} \min_{x_i \in x} \|y_j - x_i\|_2^2$. In practice, we implement the transformation to be 7-dimensional $\mathbf{T}_t \in \mathbb{R}^7$, where the first four dimensions represent a rotation quaternion and the last three dimensions are translation.

**Obtaining deformation gradient and Cauchy stress tensor.** We compute the deformation gradient $\mathbf{F}$ and the Cauchy stress tensor $\mathbf{C}$, which are necessary in the final computation of potential energy $W$ (see, *e.g.*, Equation 1 and later Equation 8), with the following equations (Kim & Eberle, 2020),

$$\mathbf{F}_t = \frac{\partial \phi(\mathbf{x}^{\text{rest}}, t)}{\partial \mathbf{x}^{\text{rest}}}, \qquad \mathbf{C} = \mathbf{F}^\top \mathbf{F}. \tag{6}$$

## 4.2 LEARNING THE UNDERLYING PHYSICAL SYSTEM VIA MINIMIZING EXTENDED ELASTICITY ENERGY

With estimation of deformation gradient and stress tensor, we can move forward to leverage our low potential energy governing principle, *i.e.*, construct total energy (Section 4.2.1) and estimate physical parameters (Section 4.2.2) by minimizing such potential energy (Section 4.2.3).

Figure 3: Simulated reconstruction sequence of multi-body point clouds scene and articulated triangle mesh.

### 4.2.1 INTRODUCING DIRECTIONAL STIFFNESS AS ANISOTROPIC NEOHOOKEAN ELASTICITY

Unfortunately, the standard classical elasticity model Equation 1 only explains a single kind of dynamic, such as soft-elastic motion with soft material, or rigid nondeforming motion with stiff material. In this section, we extend the standard formulation to general motion dynamics by introducing a new anisotropic physics quantity called *directional material stiffness*.

We propose the directional stiffness parameter, anisotropic Young's moduli $E_{\text{aniso}} \in \mathbb{R}^3_+$. Based on (Lekhnitskii et al., 1964), we introduce the following extended anisotropic Neohookean strain energy density $W_{\text{aniso}}$,

$$W_{\text{aniso},t} = \sum_{k=1}^{3} \frac{\alpha_k}{2} \left( \mathbf{a}_k^\top \mathbf{C}_t \mathbf{a}_k - 1 \right)^2, \tag{7}$$

where $\alpha_k = E_k/2(1+\nu)$ and $E_k$ denotes the $k$-th dimension of the directional stiffness parameter $E_{\text{aniso}}$. Here, for finite strains, the energy density $W_{\text{aniso}}$ incorporates invariants $\mathbf{a}_k^\top \mathbf{C} \mathbf{a}_k$ aligned with orthotropic axes $\mathbf{a}_k$, *i.e.*, $\mathbf{a}_1 = (1, 0, 0)$, and so on. We then arrive at our final total strain energy $W_{\text{total}}$, such that it combines isotropic and anisotropic contributions:

$$W_{\text{total},t} = \frac{\mu}{2} \left( \text{tr}(\mathbf{C}_t) - 3 \right) + \frac{\lambda}{2} \left( \det(\mathbf{F}_t) - 1 \right)^2 + \sum_{k=1}^{3} \frac{\alpha_k}{2} \left( \mathbf{a}_k^\top \mathbf{C}_t \mathbf{a}_k - 1 \right)^2. \tag{8}$$

Our governing principle states that the potential energy $W_{\text{total}}$ for any stable physical system should be low; therefore, we minimize this energy in our training objective (*e.g.*, Equation 10).

### 4.2.2 PARAMETERIZING ANISOTROPIC MATERIAL FIELD

To predict the material stiffness $\mathbf{E} = [E_{\text{iso}}; E_{\text{aniso}}] \in \mathbb{R}^{N \times 4}_+$, we use a neural network $f_{\theta_\mathbf{E}}$ with a set of different inputs: rest geometry $\mathbf{x}^{\text{rest}}$, spatial eigenmode weights $\mathbf{w}$, its spatial gradient $\mathbf{g}$, and potential energy $W_{\text{prev}}$ obtained in Equation 8 from the previous training epoch.

$$\mathbf{E} \leftarrow f_{\theta_\mathbf{E}} \left( \mathbf{x}^{\text{rest}}, \mathbf{w}, \mathbf{g}, W_{\text{prev}} \right). \tag{9}$$

We choose such modeling because these inputs contain useful information for estimating the extended Young's moduli $\mathbf{E}$. ① $\mathbf{x}^{\text{rest}}$: This is essential to the estimation as we are predicting per-point stiffness parameters for each point in $\mathbf{x}^{\text{rest}}$. In $f_{\theta_\mathbf{E}}$, we use cross attention layers with $\mathbf{x}^{\text{rest}}$ being query and $[\mathbf{w}, \mathbf{g}, W_{\text{prev}}]$ begin key and value to implement this per point prediction. ② $\mathbf{w}$ and $\mathbf{g}$: Typically, high stress concentrates in regions of motion variations, *e.g.*, bending or splitting. These regions are captured by spatially varying motion eigenmodes $\mathbf{w}$ and its sharp spatial variations $\mathbf{g}$, which we estimate locally over $K$ nearest neighborhood, *i.e.*, $\mathbf{g} = \arg\min_{g_i} \sum_{j \in \text{NN}_\text{K}(i)} \|(w_j - w_i) - g_i(x_j - x_i)\|_2$, where $j \in \text{NN}_\text{K}(i)$ denotes the $j$-th point in the K-nearest neighborhood for point $x_i$

Figure 4: Simulating articulated motion and predicting future dynamics with 3D gaussian splats.

with K empirically chosen as 20. ③ $W_{\text{prev}}$: The close relationship between energy $W$ and material parameter $\mathbf{E}$ can be observed from Equation 8, where $\mu, \lambda, \alpha_k$ are all functions of $\mathbf{E}$. In the beginning of training, we initialized with uniform material stiffness for the extended Young's moduli.

### 4.2.3 LEARNING PHYSICS PROPERTIES THROUGH POTENTIAL ENERGY MINIMIZATION

**A Joint Learing Objective.** The governing principle to uncover the underlying physics system is that the potential energy for any stable physical system should be low. Therefore, we optimize the underlying physical system modeled with $\theta_{\mathbf{w}}$ and $\theta_E$ by minimizing the potential energy $W_{\text{total}}$. However, as can be seen from Equation 8, simply minimizing $W_{\text{total}}$ leads the model to trivially predict zero stiffness $\mathbf{E}$. To address this, we adopt a regularization term $1/\mathbf{E}$, $\arg\min_{\theta_{\mathbf{w}}, \theta_{\mathbf{E}}, \mathbf{T}} W_{\text{total}}(\phi_{\theta_{\mathbf{w}}, \mathbf{T}}, \mathbf{E}_{\theta_{\mathbf{E}}}) + \frac{1}{\mathbf{E}_{\theta_{\mathbf{E}}}}$, which gives us a joint training objective with previously described reconstruction-based loss $\mathbf{W}_{\text{total}} = \sum_t W_{\text{total},t}(\phi_{\theta_{\mathbf{w}}, \mathbf{T}}, \mathbf{E}_{\theta_{\mathbf{E}}})$:

$$\arg\min_{\theta_{\mathbf{w}}, \theta_{\mathbf{E}}, \mathbf{T}} \mathcal{L}_{\text{recon}}(\theta_{\mathbf{w}}) + \mathcal{L}_{\text{ortho}}(\theta_{\mathbf{w}}) + \mathbf{W}_{\text{total}} + \frac{1}{\mathbf{E}_{\theta_{\mathbf{E}}}}. \tag{10}$$

**Energy Contrastive Training Scheme.** The current energy minimization objective in Equation 10 captures low-energy states of the system exhibiting observed dynamics, but it does not prevent the system from registering illegal dynamics. In physics, the reverse of this principle of low potential energy implies that unlikely dynamics should induce a high potential energy.

Following this thought, we propose a contrastive learning-based objective $W_{\text{total}}(\phi_{\theta_{\mathbf{w}}, \mathbf{T}_{\text{pos}}}, \mathbf{E}_{\theta_{\mathbf{E}}}) + 1/W_{\text{total}}(\phi_{\theta_{\mathbf{w}}, \mathbf{T}_{\text{neg}}}, \mathbf{E}_{\theta_{\mathbf{E}}})$ where $\mathbf{T}_{\text{neg}}$ represents negative transformation handles that are different from the observed data. Specifically, we sample the negative transformations $\mathbf{T}_{\text{neg}}$, where $\mathbf{T}_{\text{neg}} = \gamma \cdot \epsilon + \mathbf{T}_{\text{pos}}, \epsilon \sim \mathcal{N}(1, 0) \in \mathbb{R}^7$ and $\gamma$ controls the a noise intensity level. We take this reciprocal form since $W_{\text{total}}$ is always non-negative. In practice, we obtain $\mathbf{T}_{\text{neg}}$ by adding a Gaussian noise to $\mathbf{T}_{\text{pos}}$. Our final training loss is thus:

$$\arg\min_{\theta_{\mathbf{w}}, \theta_E, \mathbf{T}_{\text{pos}}} \mathcal{L}_{\text{recon}}(\theta_{\mathbf{w}}) + \mathcal{L}_{\text{ortho}}(\theta_{\mathbf{w}}) + \mathbf{W}_{\text{total}}(\phi_{\theta_{\mathbf{w}}, \mathbf{T}_{\text{pos}}}, \mathbf{E}_{\theta_{\mathbf{E}}}) + \frac{1}{\mathbf{W}_{\text{total}}(\phi_{\theta_{\mathbf{w}}, \mathbf{T}_{\text{neg}}}, \mathbf{E}_{\theta_{\mathbf{E}}})} + \frac{1}{\mathbf{E}_{\theta_{\mathbf{E}}}}. \tag{11}$$

Putting everything together, we arrive at **Gen-3**, a **P**otential **E**nergy **P**erspective based algorithm for **Gen**eralized dynamics **Gen**eration. We summarize the pipeline of Gen-3 in Algorithm 1, whose additional implementation and training details can be found in Section A.

## 5 EXPERIMENTS

In this section, we verify our ability to handle diverse dynamics and geometric representations. we show that our proposed framework is capable of handling heterogeneous geometry representations and diverse motion dynamics. In Section 5.2 we first test the ability to reconstruct given physical trajectory leveraging differentiability of our proposed model. We then use the predicted physics parameters to generate future frames of motion dynamics shown in Section 5.3. Finally, we provide

some qualitative experimental results using our model to generate new dynamics under new physical influences shown in Section 5.4. Additionally, to aid understanding, we a simple *two cube* experiment in Appendix Section B.1, we provide insights into the mechanism behind our method that allows for diverse interactive dynamics given the same geometry. Finally, additional experimental results, including ablation experiments, are included in Appendix.

## 5.1 EXPERIMENTAL SETUP

**Baselines.** We test our proposed method comparing to existing works, including PhysDreamer (Zhang et al., 2024b) and Simplicits (Modi et al., 2024). However, these existing works follows either linear or Neohookean elasticity using classical isotropic material stiffness. They do not innovate on the energy function itself, and thus by definition, they cannot deal with discontinuum or articulated motion with hinge or directional joints. We adopt with the following baselines:

- PhysGaussian (Xie et al., 2023) combines Material Point Method (MPM) with 3D Gaussian splatting (3DGS) to enable dynamics using predefined material parameter over the entire geometry.
- Simplicits (Modi et al., 2024) leverages deformation eigenmdoe subspace to simulate soft matter behavior using a combined linear and Neohookean elasticity energy with isotropic material stiffness. The method's only input is geometry and does not perceive motion observations. Similar to PhysGaussian (Xie et al., 2023), it uses constant material parameter for the entire geometry.
- Differentiable physics simulation, expecially differentiable MPM (DiffMPM), for material prediction is proposed by prior works through differentiable rendering (Zhang et al., 2024b) and or through constitutive law (Ma et al., 2023). Omniclaw In this comparison, we make modifications to the original method, testing directly on a 3D domain without differential rendering. This baseline method exemplifies system identification methods for isotropic Young's modulus; this reveals the necessity of anisotropic stiffness for simulation of diverse dynamics.
- SpringGauss (Zhong et al., 2024) employs a spring-mass model for 3DGS by connecting a sampled set of 3D gaussians and fitting the stiffness of springs in a differentiable simulation environment.

Many of these baseline methods work with specific geometry representation. They are not designed to tackle our task of unified dynamics generation, we make efforts to adapt them for our experiments. In cases they cannot be adapted, we use - in the table as N/A. Many efforts (Le et al., 2024; Chen et al., 2024; Liu et al., 2024b) are presented specifically for articulated dynamics. They make inherent assumptions about the simulation scheme, such as the kinematic tree. It is difficult to adapt them to generate generalized dynamics, such as soft elastic dynamics or rigid body dynamics. We therefore do not compare with these works.

**Experimental Setup.** Following (Zhong et al., 2024), we use Chamfer distance $\mathcal{D}_{\text{chamfer}}$ between the simulated prediction and the observed trajectory. We provide examples for each dynamic category.

- **Soft body dynamics**, which commonly accompanies 3D Gaussian splatting denoted as [GS] and Point Cloud denoted as [PC] in Table 2 and Table 3, under vertical gravity force. Specifically, for the rope example, we take the boundary condition of one fixed end, which is a setup for many of the baseline methods, such as SpringGauss (Zhong et al., 2024) and ConstitutiveMethods (Zhong et al., 2024; Lin et al., 2025; Ma et al., 2023; Xie et al., 2023; Zhang et al., 2024b).
- **Articulated motion**, which commonly works with meshes, denoted as [Mesh] in Table 2 and Table 3. We provide two examples: a cabinet with a rotating door and a multiple-joint robot arm. As many of the baseline methods (Zhong et al., 2024; Lin et al., 2025; Ma et al., 2023; Xie et al., 2023; Zhang et al., 2024b) do not handle mesh-type geometry. We resample the mesh and densify the point cloud on the geometry to compare with the baseline method.
- **Multi-body discontinuum**, represented as point cloud sets [PC]. Under vertical gravity, many bounded objects drop and move independently of each other.

Since our efficient dynamics simulation is fully differentiable with respect to the learnable parameters, it enables us to optimize physical parameters using differentiable simulation, as discussed in Section C.

## 5.2 RECONSTRUCTING OBSERVED DYNAMICS

We compare the reconstruction performance, first qualitatively in Figure 3. In the multi-body point cloud sequence, Gen-3 closely matches the ground truth frames at each stage, whereas other baselines

Table 2: $\mathcal{D}_{\text{chamfer}}$ of reconstruction. *Left*: soft; *Middle*: articulated; *Right*: discontinuum. *e.g.*, geometry representation cannot be used as input, physical environment not applicable, or cannot converge

| | Duck [GS] | Torus [GS] | Rope [PC] | Robot-Arm [Mesh] | Robot-Arm [PC] | Cabinet [Mesh] | Cabinet [PC] | Multi Object [PC] |
|---|---|---|---|---|---|---|---|---|
| SpringGauss | 0.498 | 0.212 | - | - | 0.836 | - | 0.910 | 0.814 |
| PhysGaussian | 0.688 | 0.145 | - | - | - | - | - | - |
| DiffMPM | 0.468 | 0.066 | - | - | 0.641 | - | 0.586 | 0.897 |
| Simplicits | 0.015 | 0.014 | 0.059 | 0.065 | 0.067 | 0.064 | 0.085 | 0.297 |
| Gen-3 | **0.013** | **0.013** | **0.046** | **0.026** | **0.016** | **0.028** | **0.038** | **0.281** |

Table 3: $\mathcal{D}_{\text{chamfer}}$ of future frame prediction. *Left*: soft; *Middle*: articulated; *Right*: discontinuum. - means N/A, *e.g.* geometry representation cannot be used as input, physical environment not applicable or cannot converge

| | Duck [GS] | Torus [GS] | Rope [PC] | Robot-Arm [Mesh] | Robot-Arm [PC] | Cabinet [Mesh] | Cabinet [PC] | Multi Object [PC] |
|---|---|---|---|---|---|---|---|---|
| SpringGauss | 2.744 | 1.451 | - | - | 1.868 | - | 0.906 | 1.595 |
| PhysGaussian | 1.192 | 0.961 | - | - | - | - | - | - |
| DiffMPM | 0.644 | 0.461 | - | - | 0.702 | - | 0.688 | 1.002 |
| Simplicits | 0.035 | 0.041 | 0.923 | 0.162 | 0.155 | 0.181 | 0.085 | 0.719 |
| Gen-3 | **0.027** | **0.011** | **0.681** | **0.070** | **0.039** | **0.085** | **0.038** | **0.173** |

cannot fall to the ground despite gravity, showing excessive stickiness and cluster collapse. This is because most prior approaches assumes single entity and do not natively handle such bounded scenarios. Likewise, in the articulated triangle mesh, our reconstructions reproduce the hinge's movement at each timestep, while other baselines cannot close the cabinet door since they treat the whole object with one single stiffness. We also show an example of the lego dozer example exhibiting articulated motion under gravity in Figure 4. We can see that our proposed approach (bottom) exhibits desired articulated motion, while other baseline approaches produce unrealisitc wobbly motion over the entire geometry domain. These indicate that our formulation better captures both rigid interactions and compliant deformations.

We then conduct quantitative comparisons in Table 2 on the reconstructed motion dynamics. Following (Zhong et al., 2024), we use the geometry distance metric to evaluate the accuracy of known simulation scenarios. Our Gen-3 consistently achieves the smallest reconstruction error through different geometry types. We also observe that our advantage over Simplicits is marginal in soft body but more significant in articulated motion, as the Simplicits and all other baseline method assumes constant isotropic material stiffness over the entire input geometry and does cannot handle cases of directional articulated motion.

### 5.3 Predicting Future Dynamical Evolution with Learned Physics Parameters

We use the first 20 frames to supervise the system to construct future dynamics prediction. We show a quantitative comparison similar to previous subsection in Table 3. In both Table 2 and Table 3, We can see that all approaches deviate from observation dynamics, but our propopsed method maintains reasonable performance across all categories of dynamics, especially in articulated and bounded continuum cases. Visualizations of soft elastic motion example can be found in Appendix Figure 10. We can see Gen-3 matches ground truth better than baselines, *e.g.* SpringGauss seems to be more bouncy than ground truth.

### 5.4 Generating New Interactive Dynamics from Motion Observation

Extending beyond reconstructing observed motion, the goal of our work is generating new dynamical behavior under new interactions or physical influences. We show qualitative results to evaluate the dynamics under new interactions in Figure 5. With one observed trajectory, where the only physical influence is vertical gravitational force, shown in the middle. We predict the physical parameters and motion eigenmodes given this observation. We put the learned system under new physical influences, where we remove the gravity and add a pulling force along the direction shown in the figure. We observe physically realistic dynamics generated by our proposed approach under the new forces.

## 6 Conclusion, Limitation, and Future Work

**Conclusion** We propose a method to model diverse dynamical behaviors through a new physical term anisotropic Young's Modulus. Gen-3 uncovers the underlying material property from motion observation inspired by the physics principle where stable physical system should be low energy state. We show that our method can not only generate diverse dynamics but also handle various geometric representations. The way our model handles bounded discontinuum is through the eigenmode weights.

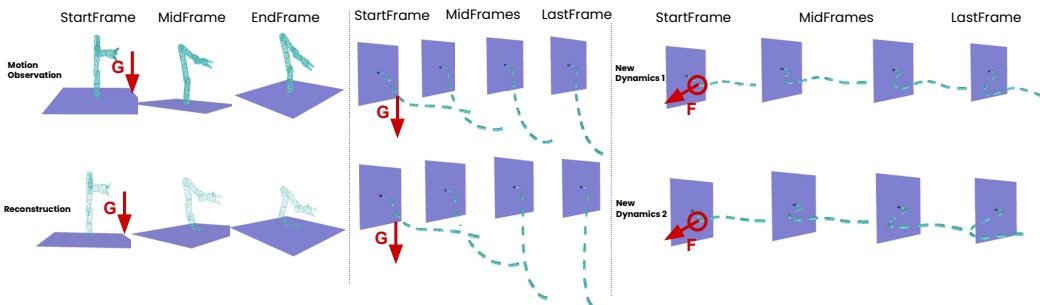

Figure 5: *Left*: reconstruction on robot arm. *Middle*: reconstruction on rope with fixed boundary condition. *Right*: new dynamic trajectories on rope under new physical influences.

When our learned motion eigenmodes weight sharply bounds the input domain, the discontinuum behavior emerges. At its core, our proposed approach enables smooth interpolation between the previously segregated domains of simulation controlled by two terms. When the learned eigenmodes become sharply bounded, behaviors of rigid body dynamics emerge. When the stiffness parameter is more anisotropic, we allow for more articulated behaviors.

**Limitation and Future Work.** A major limitation of our proposed method is that it does not inherently handle collision detection, and collision is added explicitly by setting large penalty. We also do not handle delicate thin shell, *i.e.* cloth or hair-like geometries. As our work is the first step towards unifying dynamics articulated, soft, and discontinuum with diverse geometry representations, we hope that it brings inspiration to the community to work towards these future directions.

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

# A   ADDITIONAL METHOD DETAIL AND IMPLEMENTATION

## A.1   NO OBSERVATION TRAJECTORIES

**Algorithm.** When no input observation is given, our model becomes an extended version of simplicits (Modi et al., 2024). The learned motion eigenmodes are only to minimize elasticity energy $W_{\text{total}}$ given the rest state geometry $\mathbf{x}^{\text{rest}}$.

**Experiments** In Figure 6, we can see the directional stiffness does not take effect for the case with no input observation given. In this case, our method performs similar to Simplicits (Modi et al., 2024).

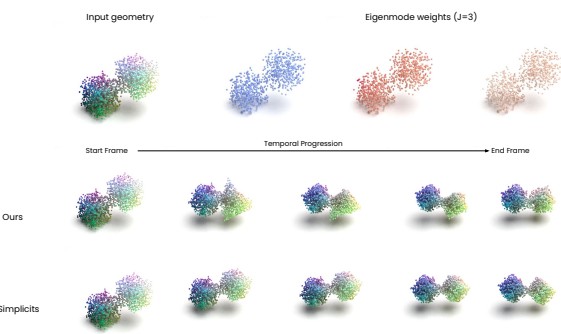

Figure 6: sample result visualization of the eigenmode weights of no input observation

## A.2   MULTI-TRAJECTORY VOTING

**Algorithm.** When multiple input observations are present, the goal is to reduce the set of possible solutions to one that aligns with all observed trajectories, $\{\mathbf{x}_{t=0}^T\}_{o=0}^O$, where $o$ denotes one trajectory. The deformation reconstruction loss now seeks to find *one common* set of eigenmode weights that simultaneously respects all observed trajectories while fitting the trajectory-wise transformations $\mathbf{T} \in \mathbb{R}^{O \times J \times 7}$,

$$\mathcal{L}_{\text{recon-multiple}}(\theta_{\mathbf{w}}, \mathbf{T}) = \sum_o \sum_t \mathcal{D}(\mathbf{x}_t, f(\mathbf{x}^{\text{rest}}; \theta_{\mathbf{w}})\mathbf{T}_{t,o}\mathbf{x}^{\text{rest}}). \qquad (12)$$

**Experiments.** In Figure 7, we show experiments with rest geometry at top, and train motion eigenmode with 1 and 3 different trajectories. The single trajectory weight is ambiguous, while multi-trajecotry voting resulting weight becomes more aligned to the actual parts of object.

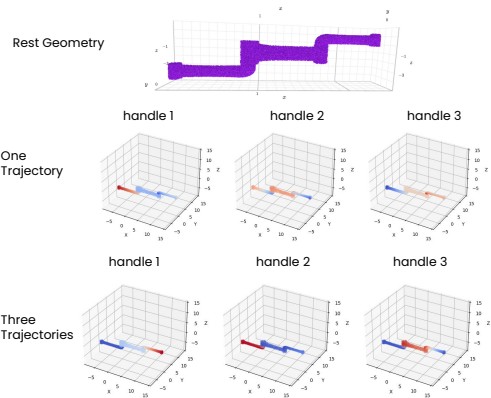

Figure 7: sample result visualization of the eigenmode weights of multiple input trajectories.

### A.3 ARCHITECTURE DETAILS

**Deformation Eigenmode Network** employs a 6-layer MLP with ELU activation and a hidden dimension of 64.

**Material Field Network** employs a multi-layer attention architecture to learn spatial field representations from 3D point coordinates and associated handle weights. The model first computes local geometric features including handle weight gradients via k-nearest neighbor least-squares fitting and local variance statistics within k-NN neighborhoods (k=20 by default). These features are concatenated with the original handle weights, elastic energy terms, and transformed handle representations to form a rich input feature vector of dimension "num-handles × 6 + 12". The core network utilizes a multilayer Attention MLP consisting of multiple local global attention layer blocks, where each layer combines local k-NN attention with global self-attention mechanisms, enabling the model to capture both fine-grained local geometric patterns and long-range spatial dependencies. The architecture concludes with a fully connected layer that maps the learned representations to the desired output dimensions, making it suitable for predicting spatially-varying material properties or deformation fields in physics-based simulations.

### A.4 TRAINING SCHEME

**Regarding contrastive training.** For the contrastive training in Section 4.2.3, we introduce a scheduled negative transformation by adding random noisy transformations to the positive transformation $\mathbf{T}_{\text{pos}}$ with a scheduling coefficient term $\alpha_e$,

$$\mathbf{T}_{\text{neg}} = \alpha_e \cdot \epsilon + \mathbf{T}_{\text{pos}}, \epsilon \sim \mathcal{N}(1,0) \in \mathbb{R}^7, \tag{13}$$

where $\alpha_e = 1.0 - \gamma^{e/T}$, $e$ is the current epoch number, $T$ is the number of total training epochs, and $\gamma \in [0, 1.0]$ is the noise reduction intensity.

As the Young's Modulus field is a function of eigenmode field deformation gradient $\mathbf{F}$, we adopt a two-stage training scheme where we first train the eigenmode field $\theta_{\mathbf{w}}$ with $\mathcal{L}_{\text{recon}} + \mathcal{L}_{\text{ortho}}$ and then we train the entire pipeline together with $\mathcal{L}_{\text{total}}$. When no observed trajectory is given, we directly train the entire pipeline without the reconstruction term $\mathcal{L}_{\text{recon}}$, and the elasticity energy loss term reduces to randomly sampled transformation $W_\epsilon$.

**Two stage training.** As the Young's Modulus field is a function of eigenmode field deformation gradient $\mathbf{F}$, we adopt a two-stage training scheme where we first train the eigenmode field $\theta_{\mathbf{w}}$ with $\mathcal{L}_{\text{recon}} + \mathcal{L}_{\text{ortho}}$ and then we train the entire pipeline together with $\mathcal{L}_{\text{total}}$. When no observed trajectory is given, we directly train the entire pipeline without the reconstruction term $\mathcal{L}_{\text{recon}}$, and the elasticity energy loss term reduces to randomly sampled transformation $W_\epsilon$.

### A.5 EXPERIMENTAL DETAILS

We used the Adam optimizer with a learning rate of $1 \times 10^{-3}$. The loss function Equation 10 is a weighted combination of several terms: a reconstruction loss weighted by $1 \times 10^3$, an orthogonality regularization term weighted by 0.1, a total energy term weighted by 1.0, and a (one over) Young's Moduli regularization term weighted by $1 \times 10^2$. These weights were chosen to balance the relative magnitudes of each loss component during training. We conduct our experiments on NVIDIA RTX 3090 GPU and RTX 8000 GPU, depending on the size of the data.

## B ADDITIONAL EXPERIMENTS

### B.1 TWO CUBE EXPERIMENT

We show that our proposed Neohookean framework with anisotropic Young's modulus can simulate different dynamics from the same initial configuration under the influence of a gravitational force, as shown in Fig. 8. The experiment is conducted with a simple sample of a geometry domain of two connected cubes. Upon initialization, the two cubes are connected at the corner, and we use two given trajectories, one with the top cube rotating around the up-axis at the origin/corner, and another trajectory of one cube translating away from the bottom cube. These two sample trajectories

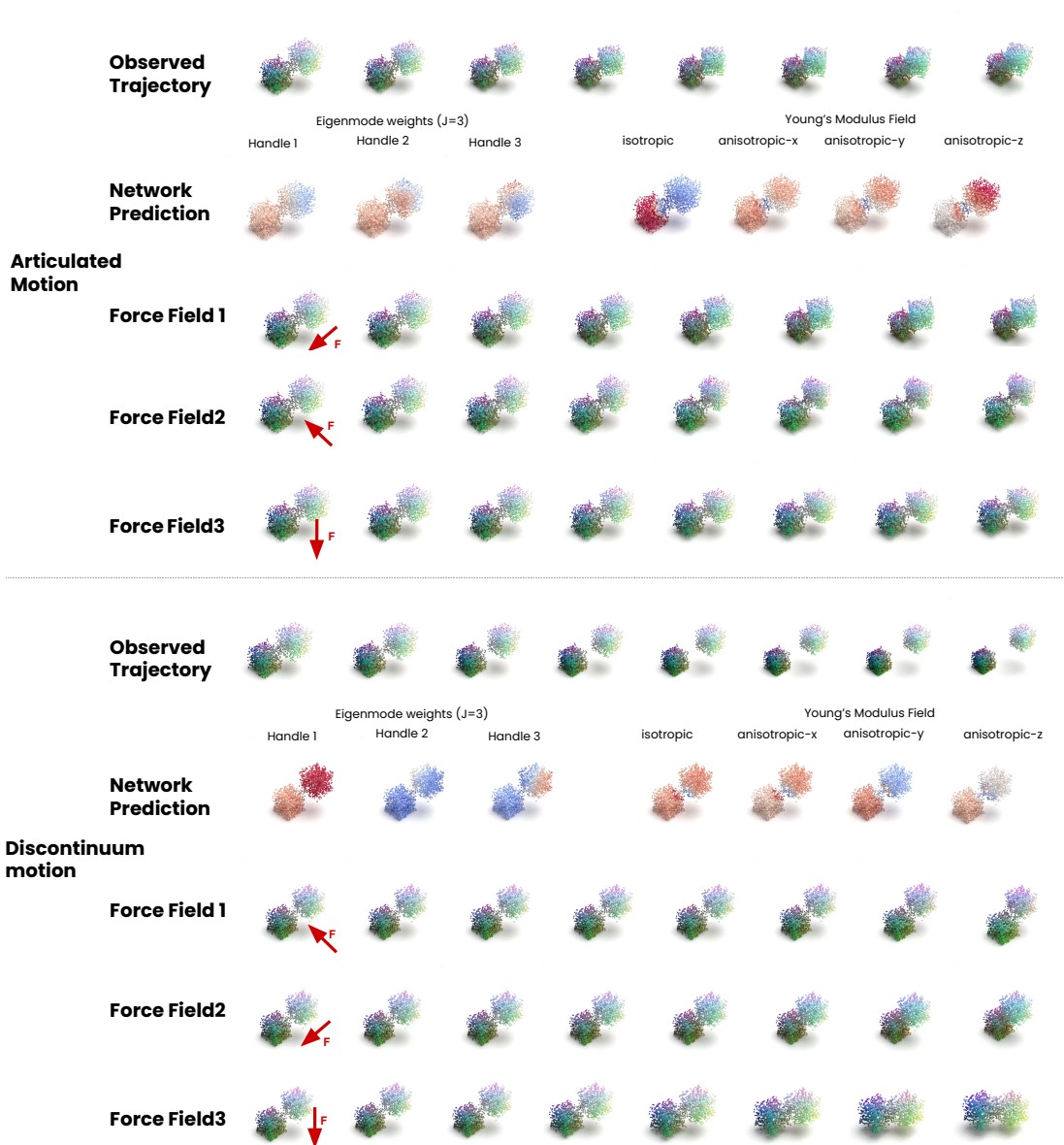

Figure 8: Results using our anisotropic elasticity framework for different systems: articulated and discontinuum mechanics. The first three rows shows the network predictions for various input system types. The bottom three sections show simulated motion trajectories under continuous force. The input observations are shown in the observed trajectory section. No observation trajectory is given to softbody.

exemplify two systems: articulated mechanics and discontinuum mechanics, which is commonly seen as rigid body simulation.

We can see from the simulation trajectories in the bottom nine rows the same governing system can present distinctive behaviors. Under the influence of gravitational continuous force in the x or y direction, the articulated system, depicted in the middle rows, presents rotational dynamics in the x-y plane, whereas under vertical (up axis) gravitational force, the articulated system almost stays in place. This happens because the anisotropic Young's Modulus field is more rigid in the vertical z-axis, as presented in the top three rows for articulated shape shown in Fig. 8. The discontinuum system, on the other hand, presents a dropping motion and moves in the direction of the force, under the same influence, as shown in the bottom rows of the simulated motion trajectories. This behavior is enabled by sharply bounded deformation eigenmodes weights, as shown in the eigenmode field section on the third row of Fig. 8. When no input trajectory is given, our proposed framework reduces to one that is similar to Simplicits (Modi et al., 2024), and the framework achieves softbody simulation in all force directions. In summary, this simple experiment demonstrates the universality of our proposed framework; with the anisotropic Young's modulus field, we are able to simulate a wide range of mechanical behaviors.

## B.2 ABLATION EXPERIMENTS

**Number of Eigenmode Handles** $J$ We compare number of eigenmode handles vs. *Motion reconstruction error* in Table 4. Note that this $\mathcal{D}_{\text{chamfer}}$ comes from fitting the motion eigenmodes, and is different from the simulation error $\mathcal{D}_{\text{chamfer}}$. We can see from Table 4 that for more complex and delicate motion needs more eigenmodes to express. For articulated and rigid motion, the fitting error levels off when $J \geq$ number of segments.

Table 4: $\mathcal{D}_{\text{chamfer}}$ of reconstruction. Lower is better.

| number of handles | $J = 2$ | $J = 3$ | $J = 5$ | $J = 8$ | $J = 10$ |
|---|---|---|---|---|---|
| Rope | 0.083 | 0.057 | 0.011 | 0.006 | 0.002 |
| Cabinet | 0.010 | 0.003 | 0.001 | 0.001 | 0.000 |
| Multi (5) Object | 0.042 | 0.016 | 0.002 | 0.001 | 0.001 |

**Neural Stiffness Field Architecture** We compare the use of various factors used in the material attention field in Table. 5. Specifically, No $\mathbf{w}, \mathbf{g}, W_{\text{prev}}$ means no attention module is applied. This experiment shows the necessity of each of the influencing factor of material field network $\mathbf{w}, \mathbf{g}, W_{\text{prev}}$.

Table 5: $\mathcal{D}_{\text{chamfer}}$ of reconstruction. Lower is better.

| | Torus | Cabinet |
|---|---|---|
| No $\mathbf{w}, \mathbf{g}, W_{\text{prev}}$ | 0.104 | 0.244 |
| No $\mathbf{w}$ | 0.033 | 0.066 |
| No $\mathbf{g}$ | 0.027 | 0.41 |
| No $W_{\text{prev}}$ | 0.040 | 0.093 |
| Full | 0.013 | 0.028 |

## B.3 ADDITIONAL RESULTS AND DISCUSSION

We show additional visualization of results in Fig. 9. The top three rows compares articulated gaussian splats between various level of ablation of our proposed method. When given no input observation, simplicits (Modi et al., 2024) creates unrealistic motion with wobbly motion of the dozer carriage. The middle row shows our method with motion reconstruction and isotropic material field. We can see that the motion reconstruction successfully removes wobbly motion of the dozer carriage. During simulation the dozer blade does not have directional motion constraint. On the bottom, we show ours

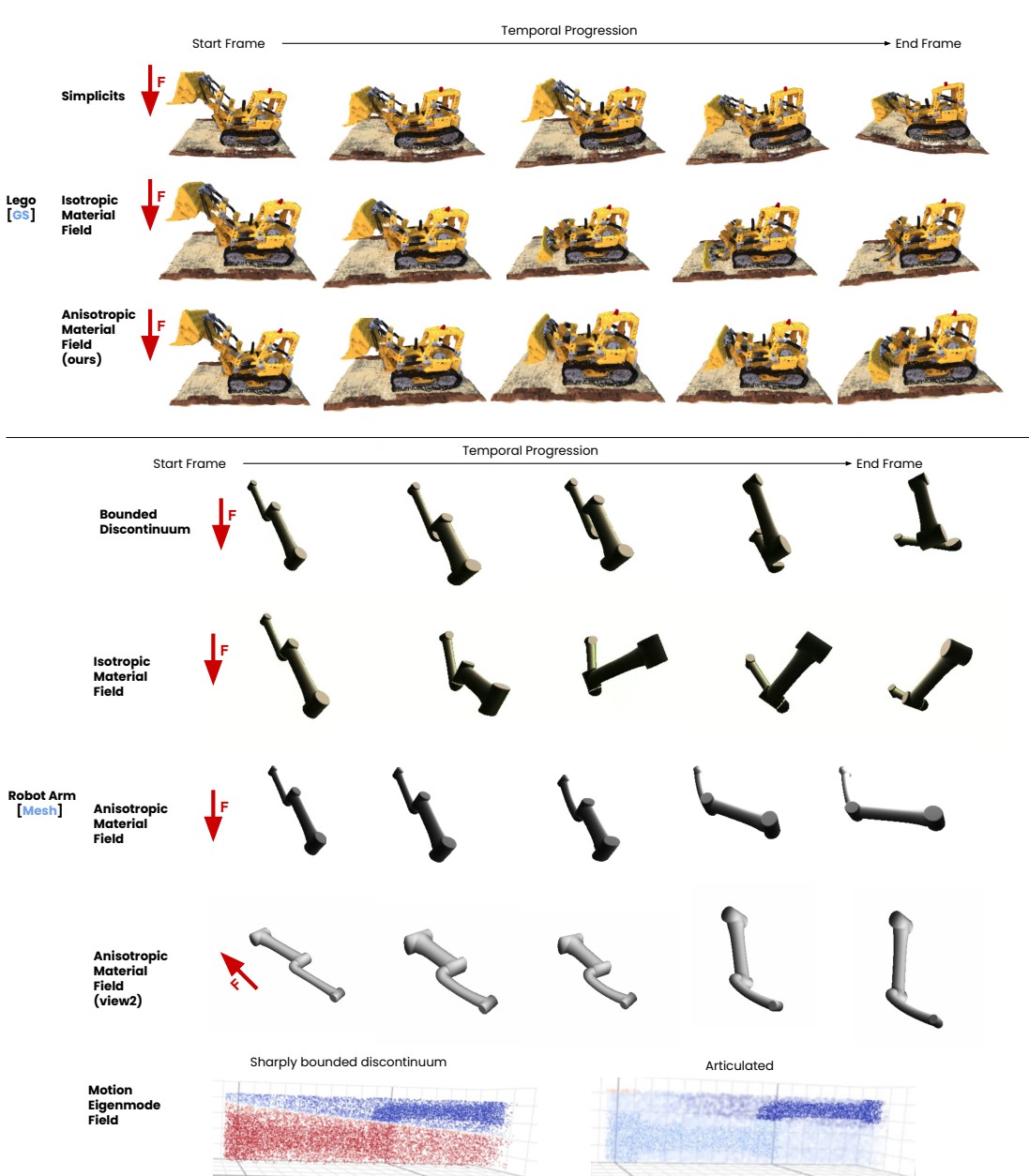

Figure 9: Various results on articulated GS of lego dozer and articulated mesh of robot arm.

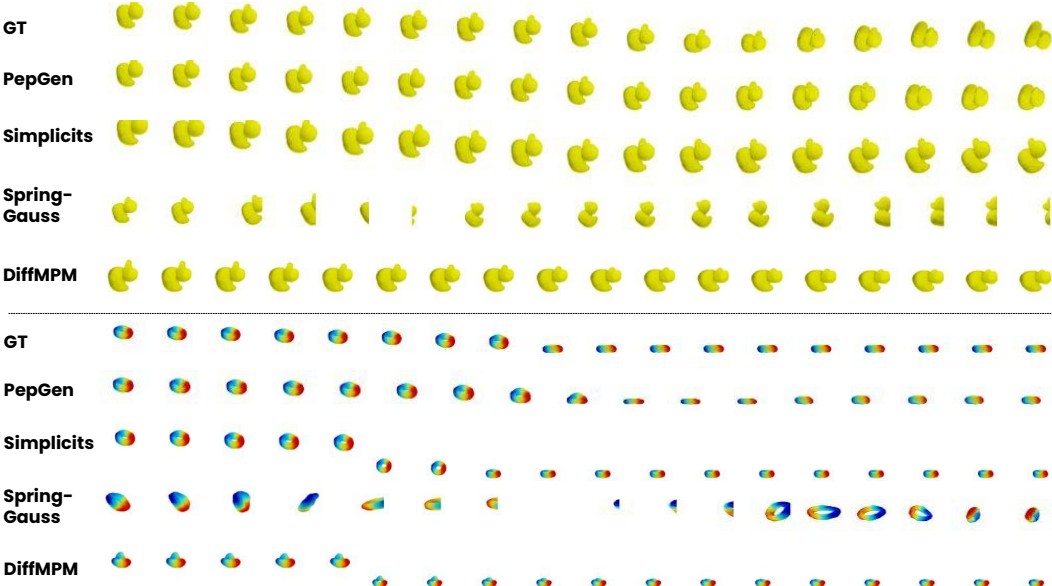

Figure 10: Reconstructing soft elastic motion and predicting future dynamics with 3D gaussian splats.

full method with both motion reconstruction and anisotropic material field. The directional motion of the dozer blade immerse and necessitate both motion eigenmodes and directional stiffness.

The bottom four rows show how our method generating diverse motion of a robot arm. When eigenmode weight sharply bounds the two geometric parts of the arm (bottom row left), the two geometric parts of the robot arm separates into two pieces. With isotropic material field, we can observe that the robot arm moves in a universal rotating joint manner. With anisotropic material field, the robot arm can move viarestricted motion direction. Additional visualization can be found here.

## B.4 ADDITIONAL COMPARISON WITH BASELINE METHOD

In our paper, we summarize differentiable material optimization methods, including Omni-PhysGS (Lin et al., 2025), NCLAw (Ma et al., 2023), and PhysDreamer (Zhang et al., 2024b), that use differentiable MPM as DiffMPM. OmniPhysGS (Lin et al., 2025) combines NCLAw (Ma et al., 2023) and PhysDreamer (Zhang et al., 2024b). Specifically, in order to compare over a more diverse forms of 3D data, we modify OmniPhysGS (Lin et al., 2025) to take in a more general form of 3D particle data (temporally-tracked point clouds), similar to the input to the original Neural Constitutive Law (Ma et al., 2023). The modification replaces the pixel-level loss and directly uses L2 point displacements as supervision signal. We name this modified version of OmniPhysGS as DiffMPM. Here we show some additional results using the original form of input (gaussian splats) and loss function(pixel level loss), of OmniPhysGS (Lin et al., 2025) and PhysDreamer (Zhang et al., 2024b). And the results are reflected in Table. 6 We can see from the table that these methods perform similar to the DiffMPM method in Table 2 for gaussian splats.

Table 6: $\mathcal{D}_{\text{chamfer}}$ of reconstruction. *Left*: soft; *Middle*: articulated; *Right*: discontinuum. *e.g.*, geometry representation cannot be used as input, physical environment not applicable, or cannot converge

| | Duck [GS] | Torus [GS] | Rope [PC] | Robot-Arm [Mesh] | Robot-Arm [PC] | Cabinet [Mesh] | Cabinet [PC] | Multi Object [PC] |
|---|---|---|---|---|---|---|---|---|
| PhysDreamer | 0.512 | 0.189 | - | - | - | - | - | - |
| OmniPhysGS | 0.466 | 0.079 | - | - | - | - | - | - |
| DiffMPM | 0.468 | 0.066 | - | - | 0.641 | - | 0.586 | 0.897 |
| Gen-3 | **0.013** | **0.013** | **0.046** | **0.026** | **0.016** | **0.028** | **0.038** | **0.281** |

## C  SIMULATION OF INTERACTION

As our goal is to extrapolate dynamical behavior under new physical influence, we derive and describe the simulation steps under our anisotropic Neohookean elasticity energy. Simulating anisotropic Neo-Hookean elasticity requires solving nonlinear equilibrium equations to compute deformations under applied forces. Newton's method is employed to iteratively minimize the total potential energy, which includes both isotropic and anisotropic contributions. Given the strain energy density $W_{\text{total}}$, the equilibrium condition is:

$$\mathbf{R}(\mathbf{u}) = \mathbf{f}_{\text{ext}} - \mathbf{f}_{\text{int}}(\mathbf{u}) = \mathbf{0}, \tag{14}$$

where $\mathbf{u}$ is the displacement field, $\mathbf{f}_{\text{ext}}$ is the external force, and $\mathbf{f}_{\text{int}}(\mathbf{u})$ is the internal force derived from the strain energy gradient.

**Newton's Method** Newton's method iteratively updates the displacement field $\mathbf{u}$ by solving the linearized system:

$$\mathbf{K}(\mathbf{u}^k)\Delta\mathbf{u}^{k+1} = -\mathbf{R}(\mathbf{u}^k), \tag{15}$$

where $\mathbf{K}(\mathbf{u}^k)$ is the tangent stiffness matrix (Hessian of the strain energy), and $\Delta\mathbf{u}^{k+1}$ is the displacement update. The internal force $\mathbf{f}_{\text{int}}$ and stiffness matrix $\mathbf{K}$ are computed as:

$$\mathbf{f}_{\text{int}} = \frac{\partial W_{\text{total}}}{\partial\mathbf{u}}, \quad \mathbf{K} = \frac{\partial^2 W_{\text{total}}}{\partial\mathbf{u}\otimes\partial\mathbf{u}}. \tag{16}$$

**Anisotropic Jacobian and Hessian** For anisotropic Neo-Hookean elasticity, the internal force and Hessian include contributions from both isotropic and anisotropic terms. The anisotropic Jacobian is

$$\mathbf{J}_{\text{aniso}} = \frac{\partial W_{\text{aniso}}}{\partial\mathbf{F}} = \sum_{k=1}^{3} 2\alpha_k\left(\mathbf{a}_k^\top\mathbf{C}\mathbf{a}_k - 1\right)\mathbf{F}(\mathbf{a}_k\otimes\mathbf{a}_k). \tag{17}$$

The anisotropic Hessian $\mathbf{H}_{\text{aniso}}$ is derived as:

$$\mathbf{H}_{\text{aniso}} = \sum_{k=1}^{3}[4\alpha_k\left(\mathbf{F}(\mathbf{a}_k\otimes\mathbf{a}_k)\otimes\mathbf{F}(\mathbf{a}_k\otimes\mathbf{a}_k)\right) + 2\alpha_k(I_4^{(k)} - 1)\mathbf{I}\otimes(\mathbf{a}_k\otimes\mathbf{a}_k)] \tag{18}$$

Finally, the total Hessian $\mathbf{K} = \mathbf{H}_{\text{iso}} + \mathbf{H}_{\text{aniso}}$ combines isotropic and anisotropic contributions. Details are included in Appendix.

## D  BROADER IMPACTS

Our work unifies rigid body, articulated body, and soft body dynamics within a single simulation framework, has significant potential for broader societal impacts. Positively, it could revolutionize fields reliant on realistic physical simulations, such as robotics, enabling the training of more versatile and adaptable robotic agents in complex environments. It also offers a powerful foundation for creating more interactive and physically accurate virtual environments for applications like augmented and virtual reality, potentially enhancing training simulations, design processes, and entertainment. However, with the increased realism and versatility of simulations comes potential negative impacts. The ability to generate highly realistic dynamic behaviors could be misused to create deceptive or harmful content, such as deepfakes involving complex physical interactions. Additionally, the development and deployment of such sophisticated simulation tools might require significant computational resources, potentially exacerbating the digital divide and concentrating power in entities with access to such infrastructure. Addressing these potential negative consequences through responsible development and deployment practices will be crucial as this technology advances.

