# OpenReview forum: "Generalized Representation for Generalized Dynamics Generation"
_ICLR.cc/2026/Conference — Submitted to ICLR 2026_

### Official Review · Reviewer_fETR · 2025-10-26

**Soundness:** 3
**Presentation:** 3
**Contribution:** 3
**Rating:** 6
**Confidence:** 4

**Summary:**

The paper proposes Gen-3, a unified, geometry-agnostic dynamics framework that extends classical elastodynamics with directional (anisotropic) stiffness, aiming to cover soft, rigid, articulated, and discontinuous behaviors within one potential-energy–based formulation. It learns a deformation field (reduced eigenmodes) and a per-point material field, then simulates under new forces. Experiments show reconstruction and short-horizon prediction across heterogeneous geometry types. Key limitations are the lack of direct comparisons to articulated-body reconstruction methods and the evaluation’s reliance on geometry metrics.

**Strengths:**

The directional Young’s moduli extend Neo-Hookean energy and enable a single system to mimic rigid/articulated/soft behaviors.

Works across meshes/point clouds/3DGS; reduced eigenmodes + neural material field is elegant and practical.

The energy-minimization with contrastive negatives is well-motivated and leads to robust reconstructions.

**Weaknesses:**

The paper convincingly demonstrates generalization across diverse object types; however, it lacks direct comparisons to specialized articulated reconstruction methods. I recommend adding a comparison with ”ArtGS:3D Gaussian Splatting for Interactive Visual-Physical Modeling and Manipulation of Articulated Objects”(whose examples indicate it can handle rigid/multi-part articulated motion) to more robustly support the claims on articulated scenarios.

**Questions:**

It would further strengthen the paper to include a comparative discussion with “Stable Constrained Dynamics”, since both works aim to build a unified simulator for elastic and rigid behaviors.

---

> ### Author Response · Authors · 2025-11-22
> **Rebuttal for fETR**
>
> **Comparison with ArtGS**
>
> - Thank you for the recommendation, we have looked into the suggested work, it seems very interesting. However, as it is arXived in July and do not have opensourced code release. We also looked into the paper, it does not seem to include the implementation details. Thus it would be difficult for us to replicate this work. Instead, we **added** discussion to explain the difference between our proposed work and the suggested baseline work. We would like to emphasize on two key differences:
>     - We aim to provide a unified framework for diverse dynamical behaviors, ranging from articulated to elastic soft body dynamics, and rigid body dynamics.
>     - Our work aim to relax as much human introduced priors as possible, including furniture templates, and predefined joint types.
>
>
> **Discuss difference to SCD**
>
> - Thank you for this helpful pointer. We have added discussion on differences to SCD in our related work section.
> - Specifically, our work and SCD focus on different scope. SCD is a numerical integration / solver formulation for simulation, assuming the material model and geometry are already given. In contrast, Our proposed work is a learning-based system identification framework that:
>
>   - infers a neural stiffness field (including anisotropic Young’s modulus) and neural deformation eigenmodes *from observed motion*;
>   - is geometry-agnostic, operating uniformly over meshes, implicit SDFs, point clouds, and 3D Gaussian splats; and
>   - uses these learned fields to create a single simulatable asset that can reproduce soft, near-rigid, articulated, and even apparently discontinuous behaviors via spatially varying stiffness, all under one potential-energy formulation (Sec. 4).
>
>   In other words, SCD unifies elastic and constrained models at the solver level for a given object, whereas Gen-3 unifies representation and material modeling across dynamics regimes and geometry types, and does so from visual data.

---

### Official Review · Reviewer_4K5Q · 2025-10-30

**Soundness:** 3
**Presentation:** 1
**Contribution:** 2
**Rating:** 4
**Confidence:** 5

**Summary:**

This paper introduces Gen-3, a framework that unifies rigid body, articulated body, and soft body dynamics through a generalized representation based on potential energy minimization. The core technical contribution lies in extending classical elastodynamics with directional stiffness via anisotropic Young's modulus, enabling the modeling of diverse physical behaviors within a single framework. Experiments demonstrate the framework's ability to handle different geometry representations and simulate various dynamics types.

**Strengths:**

- I believe this paper tackles an important problem: unifying disparate simulation paradigms (soft, rigid, articulated) within a single framework. This addresses a fundamental challenge in physical simulation with direct applications to robotics and virtual environments.

- Extending elastodynamics with directional Young's modulus may be an elegant mechanism for capturing diverse physical behaviors.

- The governing principle that stable physical systems maintain low potential energy states provides a solid theoretical foundation.

**Weaknesses:**

- There are very few visual results and no videos. Although the paper claims an anonymous page, I cannot find it.
There is no real example.

- All test examples are synthetic. It would be much more convincing to have results on real data, such as the dataset introduced in SpringGaus (and its follow-ups). Therefore the results are not convincing to computer vision people.

Minor: A related work is WonderPlay (https://kyleleey.github.io/WonderPlay/) which also tackles the problem of simulating diverse types of dynamics within a single framework, although its problem setting is single image-based and thus different from this work. The related work section may benefit from discussing the relation to it.

**Questions:**

See the major weaknesses above. I'm confused that the paper says there is a link to video results, but I cannot find it. I cannot find any video in the supplementary zip, either. Without videos, there is no way to judge if the simulation looks good or not.

---

> ### Author Response · Authors · 2025-11-22
> **Rebuttal for 4K5Q**
>
> Thank you for the review and the insightful comments.
>
> **Page Link**
> - We apologize for the, the link to a anonymous page is and **fixed link** to the
> - https://sites.google.com/view/iclr26anonymoussubmission/home?authuser=3
> - The original link is only mentioned at the end of section B.3.
>
> **Spring Gauss Data**
> - We did compare with SpringGaus on SpringGaus data in appendix, we excluded background rendering in the paper
> - We are have added more real examples from SpringGauss, excluding the background rendering.
> - We kindly disagree with the claim on visual results are not convincing to computer vision people. Our method mainly focus on providing a unified framework that is able to simulate diverse behaviors on diverse 3D representation. To the best of our knowledge, we are the first work that tackles and achieves this with the learning perspective. We believe the visual quality of our work is comparable to other ICLR or NeurIPS works such as OmniPhysGS and PhysGS. We agree with the room of improvement of our work's visualization without background rendering and etc, but we still believe the merits of our work is shown through the variety of visual examples of diverse dynamics generated provided.
>
>
> **Related Work with WonderPlay**
> - We have added discussion with related work with WonderPlay
> - Recent works including WonderPlay uses keypoint trajectory and inductive priors to determine appropriate physics simulator to use to simulate behaviors in a 2D scene. Our work pursues a completely different philosophy. By removing assumptions and priors, we aim to directly learn the underlying physical system the from observed behaviors in a unified physical system. In this way, we are freed from limitations and mismatches of physical priors.

---

### Official Review · Reviewer_tim6 · 2025-11-01

**Soundness:** 2
**Presentation:** 2
**Contribution:** 2
**Rating:** 4
**Confidence:** 4

**Summary:**

This paper targets dynamics generation for objects across diverse material types. The method learns dynamics via neural deformation eigenmodes coupled with transformation handles, while a learned material field captures material-specific attributes. Experiments show superior performance across multiple domains.

**Strengths:**

* The method is evaluated against multiple baselines and demonstrates superior performance.
* The introduced directional stiffness parameters enable modeling of a wide range of materials.

**Weaknesses:**

1. Clarify $W_{total}$ in Eq. 10. The definition of $W_{total}$ is unclear. Does it denote the sum of energies over all frames/observations, or an energy for only one frame?
2. Generalization beyond observed time. The method reads primarily as reconstruction from observed data. How does the model predict dynamics beyond the observation window? From Eq. (2), only the transformation handlers appear explicitly time-related. Please explain:
    1. How handlers reconstructed from observations extrapolate to unseen dynamics;
    2. Whether there is an explicit dynamic prior or evolution rule for handlers/eigenmodes;
    3. Any rollout strategy or error-accumulation analysis for long-horizon predictions;
    4. An efficiency analysis of inference speed.
3. The paper should include the discussion of [1,2] in related work, which are also related to 4D generation and dynamic simulations.

[1]. Cao, et al. Neural Material Adaptor for Visual Grounding of Intrinsic Dynamics. NeurIPS 2024
[2]. Shao, et al. GausSim: Foreseeing Reality by Gaussian Simulator for Elastic Objects. ICCV 2025

**Questions:**

Please refer to weaknesses.

---

> ### Author Response · Authors · 2025-11-22
> **Rebuttal for Reviewer tim6**
>
> Thank you for the review and the insightful comments.
>
> **$W_{total}$**
> - Thank you for this question, $W_{total}$ is defined for every frame with respect to the rest state, where the timestep t is introduced when calculating the Cauchy stress tensor. During training, we sum over energies over across all frames,
> - To make this more clear, we edited the equation to include per frame timestep t in the equations and losses
>
>
> **Generalization beyond Observed time**
>
> - How handlers reconstructed from observations extrapolate to unseen dynamics;
>     - This question relates to how simulation works.  During training, our method works to identify the physical material parameters: given a observed window of dynamics:
>         - deformation eigenmodes $w(x)$,
>         - material field $E(x)$,
>         - and per-frame handle transformations $T_t$.
>
>          Eq. (2–5) and Algorithm 1 describe this reconstruction stage: we use $T_t$ as latent generalized coordinate that parameterize the observed deformations $\phi_{\theta_w,T_t}(x_{\text{rest}})$ over time.
>
>   - Beyond the observation window, we do not continue to optimize or regressing $T_t$ from images. Instead, we use the identified eigenmodes $w(x)$ and stiffness field $E(x)$ to run a standard elastodynamic simulation: $\mathbf{R}({q}(t)) + f_{\text{int}}(q(t); E) = f_{\text{ext}}(t)$, where q(t) are generalized coordinates in the reduced space spanned by the learned eigenmodes, and $f_{\text{int}}$ is derived from the potential energy. This is the standard second-order ODE obtained from the energy model, as described in Appendix C. We use the last observed state (positions/velocities) as initial conditions and integrate forward under gravity and any specified external forces.
>
>
> - Whether there is an explicit dynamic prior or evolution rule for handlers/eigenmodes;
>     - In our work, we do not use dynamics prior or explicit rules to define transformation handles, or rules to allow for more . However, they exists as prior works that rely on categorical priors or using predefined joint types to do articulated shape simulation.
> - Any rollout strategy or error-accumulation analysis for long-horizon predictions;
>     - We follow the evaluation setup of SpringGauss: after using an initial portion of the trajectory for system identification, we roll out the simulator for future frames and measure Chamfer distance between simulated states and ground truth. Table 3 reports these future-frame $D_{\text{Chamfer}}$ errors across all scenes.
>      - We also add a plot of error vs. rollout horizon in the Appendix section.
>
> **Related Works**
>   - Thank you for pointing out these related works. We have added discussion in the related work section.
>   - Specifically, NeuMA leverages differentiable MPM for identifying system parameters. Similar to NCLaw, and Omniphys4D (as compared in the paper), it uses elasticity and plasticity law to regress the parameters to generate desired motion. GausSim proposes a hierarical grouping in the Gaussian space to speed up simulation. However, these works only considers soft elastic objects, assuming consistent soft material for the simulated shape. Our proposed method, on the other hand, goes beyond standard elastodynamics by introducing a new physical parameter, directional stiffness (anisotropic Young’s modulus). This additional term enables us to capture a wide spectrum of dynamic behaviors, ranging from soft elastic materials to rigid bodies and articulated systems.

---

### Official Review · Reviewer_LvtX · 2025-11-05

**Soundness:** 3
**Presentation:** 3
**Contribution:** 3
**Rating:** 6
**Confidence:** 3

**Summary:**

The paper proposes e a unified framework that integrates different physical systems, including rigid body, articulated body, and soft body systems.
The framework takes the potential energy perspective that the potential energy for any stable physical system should be low. With this perspective the problem can be cast into elastodynamics, using the elasticity energy function to enable interaction.

The framework contains two learnable networks: theta_W which learns the motion eigenmode weights, and theta_E which learns Young’s Moduli E. The learned components from the networks can then be used to simulate motion dynamics. For training, in addition to the reconstruction loss between predicted and observed outputs, the paper introduces orthogonality regularization (for theta_W) and the strain energy (for theta_E) in the loss term.

Experiments include evaluations on soft-body dynamics, articulated motion, and multi-body discontinuum systems. The proposed method is compared against several methods, including PhysGaussian, Simplicits, the differentiable MPM method, and SpringGauss. Results show that the proposed approach achieves lower reconstruction errors than the baselines and demonstrates strong ability in long-term dynamics prediction.

**Strengths:**

* The proposed framework can handle a wide range of physical systems (soft, rigid, and articulated), whereas many existing methods are limited to one or a few specific types.

* The experiments cover multiple diverse 3D scenes to better demonstrate the ability of the proposed method to handle different physical systems.

**Weaknesses:**

I have some additional questions listed in the section below.

**Questions:**

- The paper mentions “More visual results on our anonymous page”, but I was not able to find a link to the anonymous page from the paper and the appendix…
- Can the model generalize to unseen scenarios? For instance, objects with the same material properties but different shapes and/or different numbers of input nodes, or the same learned objects but interacting with different fixed boundaries?
- It might be helpful to also demonstrate how each component of the loss function contributes to training performance, such as,, the impact of incorporating potential energy in the loss, and the effect of introducing negative transformation handles in the energy loss.

---

> ### Author Response · Authors · 2025-11-22
> **Rebuttal for Reviewer LvtX**
>
> Thank you for the review and the insightful comments.
>
> **Page Link**
> - We apologize for the, the link to a anonymous page is and **fixed link** to the
> - https://sites.google.com/view/iclr26anonymoussubmission/home?authuser=3
> - The original link is only mentioned at the end of section B.3.
>
> **Generalization**
> - We focus on generalized dynamics from sparse observation,
> - *Same shape, different number of nodes*: Yes, our method uses neural field which can work with arbitrary number of nodes as sampled as long as the shape is consistent. In Table 3 we already evaluate the same physical object (“Cabinet”) represented as both a triangle mesh and a point cloud (“Cabinet [Mesh] / [PC]”). Gen-3 achieves consistent prediction quality across these different discretizations (and for Robot-Arm [Mesh]/[PC] as well), which demonstrates geometry-representation agnosticism—the learned neural fields can be queried on arbitrary samples of the same rest shape.
> - *Same object, different boundary condition*: Yes, our method learns the neural field together with material field for the shape, and the simulator can take in the shape and any other predefined boundary condition to simulate the behavior. Examples including “two-cube” in Appendix, and also in Figure 5, pushing/pulling the same object from new directions or with new contact points not seen during training.
> - *Same material different shape*:  We do not claim cross-shape generalization from a single training object to entirely new shapes. Our focus is to unify simulation across types of dynamics and geometry representations (soft, rigid, articulated, discontinuous; SDF/tet/mesh/point cloud/Gaussians) for each individual object rather than learning a universal meta-model across shapes. That said, one can achieve new shape by using a generalized conditional geometric neural field, which is unsolved and an active line of research. We mainly tackle ways to unify and generalized dynamics instead of generalized shape reconstruction
>
>
> **Loss Components**
> - There are four loss components:
>     - 1. reconstruction loss and 2. orthogonal regularization terms are uses observed motion trajectory to group eigenmodes and transformation handles.
>     - 3. W_total, pos is the positive energy term forces the learned physical parameters to minimize the physically consistent potential energy given the eigenmodes and transformation handles.
>     - 4. W_total, neg is the negative energy term that forms  a contrastive pairing: among nearby deformations in transformation space, the observed one ais encouraged to be energetically favorable compared to its perturbations to avoid trivial solutions where all energies are uniformly low or high, and encourages local energy shaping that is sensitive to physically plausible motions.
>
> - To make this more clear we added this clarification in the rebuttal revision

---

### Author Response · Authors · 2025-11-29
**General Comments**

We would like to thank all reviewers for the insightful comments.

We acknowledge the positive feedback, including:
- Proposed method is novel and elegant (LvtX, tim6, 4K5Q, fETR)
- Address an important research question (LvtX, 4K5Q)
- Experimental results are extensive and shown to be effective (LvtX, tim6, fETR)


We would like to address the commonly raised question regarding the link to the [anonymous page](https://sites.google.com/view/iclr26anonymoussubmission/home?authuser=3)

We apologize the link does is not correctly attached on the first page, and is only mentioned one other time at the end of section B.3.
[https://sites.google.com/view/iclr26anonymoussubmission/home?authuser=3](https://sites.google.com/view/iclr26anonymoussubmission/home?authuser=3)


We hope our work on a novel formulation to unify diverse dynamics behaviors can offer valuable insights and inspiration to the research community.

---

### Meta-Review · Area_Chair_zNNQ · 2025-12-13

**Summary:**

The paper introduces PepGen (Potential Energy Perspective for Generalized dynamics Generation). Pepgen is designed to unify rigid b ody, articulated body, and soft body dynamics into a single, geometry-agnostic model, based on the principle that the potential energy of the system should be low in its stable state. The model predicts a neural field for the deformation of the object.

The main contribution of the paper is introducing a unified representation for different physical systems (soft, rigid, articulated) and geometry representations (meshes, point clouds, Gaussian splats) in a single learned framework.

Summary of reviewer's concerns:
* Missing Key Baseline Comparisons: The initial submission lacked essential comparative results against methods like SCD and Gen-3.

* Weak Soft Body Evaluation: The soft body experiments were limited and unconvincing, failing to adequately demonstrate the framework's capability to handle complex non-smooth or high-deformation soft dynamics.

* Clarity on Dynamics Unification: The paper lacked sufficient technical detail and rigorous explanation on how the single potential energy formulation successfully unifies discrete dynamics (rigid/articulated) with continuous dynamics (soft) via directional stiffness.

* Generalization to novel geometries that are significantly different from the training data

**Reviewer Concerns:**

* The main criticism of the reviewers was the lack of visual results, real examples and rendering of the videos. At ICLR we do not expect the papers to have sophisticated simulation rendering like in Computer Vision conferences (ICCV or Siggraph). However, AC agrees with reviewers that the provided examples are relatively simple. The simulations include only a single object, and there is a single example provided for each type of dynamics or input modality.

* Initial concerns were raised by the reviewers regarding the model's limited generalization across unseen shapes and its inability to extrapolate to longer time horizons. Although the authors’ rebuttal explicitly narrowed the scope by disclaiming these features, the AC views this restriction as defeating the fundamental premise of a “unified simulator” framework.

**Reviewer Scores:**

How  the reviewer would have changed their score if they had been able to participate fully in the discussion:
* Reviewer 4K5Q, Rating 4 -- unchanged
* Reviewer LvtX. Rating 6 -- unchanged
* Reviewer tim6. Rating 4 -- unchanged
* Reviewer fETR. Rating 6  -- unchanged

---

### Decision · Program_Chairs · 2026-01-26

Reject